# The Influences of Moisture on the Mechanical, Morphological and Thermogravimetric Properties of Mineral Wool Made from Basalt Glass Fibers

**DOI:** 10.3390/ma13102392

**Published:** 2020-05-22

**Authors:** Andrej Ivanič, Gregor Kravanja, Wadie Kidess, Rebeka Rudolf, Samo Lubej

**Affiliations:** 1Faculty of Civil Engineering, Transportation and Architecture, University of Maribor, Smetanova 17, 2000 Maribor, Slovenia; andrej.ivanic@um.si (A.I.); wadie.kidess@gmail.com (W.K.); 2Faculty of Chemistry and Chemical Engineering, University of Maribor, Smetanova 17, 2000 Maribor, Slovenia; 3Faculty of Mechanical Engineering, University of Maribor, Smetanova 17, 2000 Maribor, Slovenia; rebeka.rudolf@um.si

**Keywords:** mineral wool, basalt fibers, moisture effect, compressive strength, degradation, SEM-EDX, STEM, thermal stability, roofing

## Abstract

Mineral wool made from basalt fibers is frequently used as an insulating material in construction systems. In this study, both unused mineral wool and wool obtained from the softened roofing area were comprehensively analyzed in a laboratory using different characterization techniques. Firstly, the initial water content and compressive strength at 10% deformation were determined. Secondly, microstructure and surface chemical composition were analyzed by scanning electron microscopy (SEM) equipped with energy dispersive X-ray spectroscopy (EDX). To study heterogeneities near the fiber surface and to examine cross-sectional composition, a scanning transmission electron microscope (STEM) was used. Finally, to verify possible reasons for resin degradation, thermogravimetric analysis and differential scanning colometry (TGA-DSC) were simultaneously carried out. The results show that natural aging under high humidity and thermal fluctuations greatly affected the surface morphology and chemical composition of the fibrous composite. Phenol-formaldehyde and other hydrophobic compounds that protect fibers against moisture and give compressive resistance were found to be degraded.

## 1. Introduction

There are many different thermal insulation materials available for ensuring better energy efficiency of construction systems. Thermal and hygrometric performance varies with the type of insulation used, and depends on the inner material structure and composition [1,2]. Mineral wool is one of the most frequently used insulating fibrous composites in the construction industry [3]. When basalt fibers are used as the hardening phase, mineral wool has an extremely uneven and porous microstructure with good fiber/resin adhesion that provides excellent insulating performance [4]. However, as seen from building practice, temperature, moisture, and frost formation can greatly affect the thermal and mechanical properties of insulating materials that are used in flat or pitched roof boards, facades, and foundations [5].

Despite recent extensive research on the hygroscopic performance of insulating materials, there are still many uncertainties in our understanding of the durability and performance of such materials, and how moisture affects degradation mechanisms. Most studies have focused on laboratory measurements of the accelerated aging of new insulating materials under temperature and moisture conditions close to natural ones. For example, Lund and Yue [6] studied the accelerated aging effect of high humidity and water on the surface morphology and crystallization of new basalt glass fibers. They found that water greatly impacted the surface morphology and crystallization behavior of basalt fibers. Vrana and Gundmundsson [7] compared cellulose and stone wool in terms of moisture performance over a period of a few days. They found that there were minor changes in water vapor permeability. Achcaq et al. [8] tested the thermal, morphological, and structural properties of two types of glass fiber under various humidities. They concluded that the absorption of water in coated glass fibers was greatly influenced by the composition of the binders. Jiřičková and Černý et al. [9] investigated the effects of hydrophilic admixtures on the water vapor diffusion coefficient, retention curve, and liquid moisture diffusivity of mineral wool. They found that hydrophobic admixtures in mineral wool considerably enhanced liquid water transport and water storage parameters. Jerman et al. [10] observed that the thermal conductivity of mineral wool rose very quickly with increased humidity, from 0.041 W/mK in dry conditions up to 0.900 W/mK in saturated conditions. Vrana and Bjork [5] conducted an interesting study of frost formation and condensation in fibrous mineral wool with different densities. Their results showed that moisture was 2.5 or 3 times higher than expected in insulating materials with lower density.

On the other hand, only a few studies concern the performance and durability of mineral wool insulation under on-site conditions. In order to find the most suitable insulation material for new constructions, it is important to identify the actual performance and durability of the existing insulation. Rasmussen et al. [11] presented the hydrothermal performance of several insulation materials subjected to natural Danish conditions over two years. Toman et al. [12] presented an assessment of the hygrometric and thermal performance of interior hydrophilic mineral wool after four years. 

As a waste material, mineral wool is currently considered to be unrecyclable. Several solutions to the problem of improving recyclability have recently been proposed. Gutiérrez-Orrego et al. [13] studied the reinforcement of soil-cement blocks by adding mineral wool waste and sisal fibers. They observed that bending strength and compression resistance increased with the addition of mineral wool waste and sisal fiber. Väntsi et al. [14] presented new utilization options for mineral wool waste as a filler in wood polymer composites. The findings suggest that the addition of mineral wool enhanced the moisture resistance and mechanical properties of wood-based composites. 

The aim of this study was to identify durability and possible reasons for the degradation of mineral wool manufactured from basalt fibers by comparing new samples to samples extracted from a flat softened roofing area that was exposed to high humidity for approximately 10 years. Comprehensive analysis and material characterization were carried out in a laboratory. In the first part, the study presents how real-life conditions, including high humidity and cyclic thermal loads over a long period of time, affect the mechanical performance of insulation. In the second part, new insights are provided about the morphology and chemical composition of degraded samples that greatly alter the surface of basalt fibers. The third part deals with thermogravimetric and thermal transition analyses to determine the thermal stability of hydrophobic resin used in insulation material.

## 2. Materials and Methods

Globally, mineral wool made from basalt fibers is frequently used as an insulating building material. To address possible reasons for degradation, insulating samples were comprehensively analyzed in a laboratory using different characterization techniques.

### 2.1. Materials from Case Study

DDP-RT Thermal (Knauf Insulation) material samples were extracted from an 8400 m^2^ flat unvented roof of a commercial building located in Slovenia, Central Europe, characterized by a Cfb climatic zone. The roofing structure consisted of several layers: On the top was a 1.2 mm thick polyvinyl chloride with plasticizer (PVC-P) waterproofing layer; under that was a 200 mm thick thermal insulation of mineral wool made from basalt fibers with a density of 160 kg/m^3^, followed by a 0.3 mm vapor barrier and 1.25 mm thick trapezoidal profile metal sheeting anchored into steel roof beams (Figure 1). 

During the refurbishment, notable errors on the roof were detected. On the uneven roofing area, it was found that the steam barrier was not correctly installed and paired to the roof attic. Samples of mineral wool were taken from the softened roofing areas, and immediately enveloped in polyethylene and transported to the laboratory for further analysis (Figure 2).

The thermal insulation samples were made of basalt rock that originated from solidified lava. They were spun by melting under hydrostatic pressure at approximately 1450 °C into a fiber-like structure, then joined together with phenol-formaldehyde binders and silane additives (SiH_4_) to achieve high water repellency and mechanical strength. Compared to the production of wool with glass or carbon fibers, this process is simpler and cheaper, with a high rate of recyclability [15].

### 2.2. Moisture Content and Mechanical Testing

Moisture in degraded mineral wool samples was determined using the gravimetric method, according to UNI EN ISO 12570. Water content (*u*) was calculated by means of Equation (1):*u* = *m*_w_ − *m*_0_/*m*_0_ × 100(1)
where *m*_w_ and *m*_0_ are the weights of the extracted samples taken from the softened roofing area and of a sample conditioned at 105 °C in a climatic chamber until constant mass, respectively. A precision scale with gradations of 0.01 g was used to determine weights. Additionally, adsorption isotherms for new and aged samples were measured in the laboratory at 25 ± 1 °C. Dry samples were placed in a desiccator and conditioned for at least 24 h at three relative humidity levels (33% ± 2%, 56% ± 2%, and 97% ± 2%) corresponding to different aqueous saturated solutions [16]. The moisture content at different relative humidity levels was calculated according to Equation (1). 

Compressive strength at 10% deformation was determined on dry samples according to SIST EN 826 using a Zwick/Roell Z010 universal test machine [17]. Test specimens were squarely cut to dimensions of 100 × 100 mm^2^ with a thickness of 100 mm. In order to determine moisture content and compressive strength at 10% deformation, at least five samples were used for testing.

### 2.3. Morphological and Chemical Composition Characterization

The morphology and composition of basalt fibers bonded with phenol-formaldehyde resin were determined using a Quanta 200 3D ESEM (environmental scanning electron microscope) (FEI Company, Hillsboro, OR, USA) equipped with FEI Sirion 400 NC EDX (energy dispersive X-ray spectroscopy, FEI Company, Hillsboro, OR, USA). EDX was used to examine the composition of different elements on the surface of fibers. In order to study the cross-sectional microstructure composition of elements in thin basaltic fibers, a STEM (scanning transmission electron microscope, Joel, ARM, 200 CF, Japan) was used. Some of the samples were purified and washed in 95% ethanol prior to examination.

### 2.4. Thermal Characterization

Thermogravimetric analysis and thermal transition of material samples were carried out using a TGA-DSC (simultaneous thermogravimetric analysis and differential scanning calorimetry) instrument (Mettler Toledo, Columbus, Ohio, USA). Approximately 10 mg of the new or degraded material was placed in the vials and heated in an N_2_ atmosphere at a rate of 10 °C/min from 30 to 1000 °C. Mass loss caused by thermal change was measured as a consequence of water evaporation and combustion of resin and other organic compounds.

## 3. Results and Discussion

### 3.1. Water Content and Mechanical Performance

Table 1 reports the initial water content (*u*) in the aged samples and their mechanical performance. The samples had moisture content in a range from 0.53 wt.% to 4.90 wt.%. Generally, moisture in mineral wool is very low (less than 0.4 wt.%) because of the presence of phenol-formaldehyde binder and other hydrophobic compounds. In order to compare the moisture storage properties between new and aged samples, adsorption isotherms were measured (Figure 3). The new samples had values at least two times lower than the aged ones. One possible explanation could be that the undamaged hydrophobic admixtures permitted a decrease in the sorption of water into the material.

The higher presence of moisture inside the aged samples might have been caused by the degradation of phenol-formaldehyde resin and other organic compounds that are essential for the protection of insulating material against water [18]. Residual water which did not evaporate during the summer caused partial degradation of resin by depolymerization reaction (Figure 4).

Compressive strength at 10% deformation was determined for the degraded samples, and the values were compared to the new samples (etalon) with a declared value (*σ*10, dec) of 70 kPa (Figure 5). The results show that the presence of moisture in insulation over time had a high impact on the compressive strength of mineral wool made from basalt fibers. The mean compressive strength (*σ*10, mean) of a sample extracted from the softened roof area was 93% lower than the declared value. High humidity and temperature oscillations that can reach up to 70 °C on the roofing surface in the summertime have a high impact on the mechanical properties of insulating material and, consequently, on degradation. To identify the precise mechanisms that cause the loss of compressive strength, detailed microanalysis was carried out.

### 3.2. Microscopic Analysis

The surface morphology and chemical composition of the surface of degraded samples were investigated using SEM coupled with an EDX detector, and the presence of elements in the interior of the fibers was determined with STEM analysis.

#### 3.2.1. Scanning Electron Microscopy

Scanning electron microscopy was conducted to provide detailed microscopic analysis and surface morphology of samples under different magnifications (Figure 6a). SEM showed a random distribution of basalt fibers and the presence of non-fibrous melt inserts in the form of droplets (Figure 6a). Locally present melt inserts can be explained by the uneven distribution of a binder during the production procedure. Figure 6a–c represents the microstructure of new wool, showing fibers that have smooth surfaces. Figure 6d–f shows the aged samples taken from a softened roofing area, where fibers are surrounded by a uniform layer with many cracks and bulges. One of the reasons for such a noticeable change in microstructure could be the elongation (α) of insulating material under different cyclic thermal loads. The elongation of basalt fibers is about 8 × 10^−6^ K^−1^, while phenol-formaldehyde resins have a higher value ranging from 30 × 10^−6^ to 45 × 10^−6^ K^−1^ [19]. Different elongation under natural aging could cause the resin film to swell at sites with poor adhesion, and consequently, small areas without hydrophobic protection that are more sensitive to moisture could open up. Thus, we can conclude that weather conditions over a prolonged period of time have a very high impact on surface morphology.

#### 3.2.2. SEM Coupled with EDX Detector

The SEM-EDX spectra presented in Figure 7 and Figure 8 indicate that the composition on the fibrous composite surface changed after being exposed to aging under high humidity and thermal fluctuations for approximately 10 years. Figure 7 presents a micrograph of a phenol-formaldehyde layer with an average thickness of approximately 0.5 µm that uniformly surrounds the basalt fiber. Since electron beam penetration of the material was limited to 1 µm detection, elements mainly concerned the binder layer and only a certain part of the fiber. Therefore, different surface chemical composition was expected in the degraded samples. Different compositions in degraded samples were correlated with two characteristic surface defects: cracks and bulges, as shown in Figure 8. This is the reason why Si was less present in the new uniformly coated sample compared to the degraded one. Cracks and bulges enabled different electron beam penetration, as there was a lack of resin in the surface area. Pure basalt fibers without binder consist mainly of SiO_2_ and Al_2_O_3_ [20]. The detailed chemical surface composition of the new and degraded samples is presented in Table 2.

Fibrous samples with cracks and bulges on the surface showed an increase in the relative amount of Na, Mg, Al, Ca, and K (potassium) elements compared to Si (Table 3). This was similar to the findings of Lund and Yue [4], who measured relative amounts of elements compared to Si in a basalt surface which had been immersed in deionized water at 70 °C for several weeks. Since aged samples showed a relative increase of Al and Ca elements, we might assume that Al_2_O_3_ and CaO deposits were formed on the surface. A higher level of CaO in the aged samples suggests that basalt corrosion was in progress [21].

#### 3.2.3. STEM Analysis

A scanning transmission electron microscope was used to examine heterogeneities near the aged fiber surface in nano-scale (10–100 nm) under high magnifications ranging up to 300,000 times. The results show the formation of a thin ribbed layer on the basalt surface that could be a consequence of ion exchange. Additionally, with parallel electron beam penetration through the thin fibers, it was possible to determine the presence of elements on the surface and in the interior. The results indicate the presence of a wide spectrum of elements, with Si, Mg, Al, Ca, O, and Fe being the most common in the cross section perpendicular to the basalt fiber (Figure 9).

One possible factor contributing to the degradation of the basalt surface layer could be inadequate phenol-formaldehyde resin curing during production. Many manufacturers try to optimize energy consumption, and the polymerization reaction during resin curing takes a great deal of energy and time [22]. To verify the possible degradation of resin, simultaneous TGA-DSC analysis was used to determine the mass change (thermogravimetry) and thermal transition (DSC).

#### 3.2.4. Thermal Behavior

Figure 10 presents thermogravimetric and thermal transition analysis of new and aged insulating samples in the temperature range from 30 to 1000 °C. TGA shows that on both curves, weight loss was caused by heating. Weight loss between 30 and 150 °C corresponded to the evaporation of the absorbed moisture. The positive entering heat flux using differential scanning calorimetry corresponded to energy absorbance during water vaporization. Weight loss up to 800 °C corresponded to the combustion of the binder and other organic hydrophobic agents. Negative heat flow showed that energy was released by the burning of phenol-formaldehyde resin [23]. A noticeable weight difference between new and aged samples was observed after 300 °C (Figure 11). Combustion affected the aged sample to a lesser extent. This could be because some of the formaldehyde resin may have already been depleted before the analysis was carried out. The slight increase in heat flux at approximately 900 °C might correspond to exothermal transformation due to the crystallization of the samples.

## 4. Conclusions

The results of the measurements reaffirmed the sensitivity of mineral wool from basalt fibers used as thermal insulation material. The main conclusions are:Degraded insulating materials extracted from the softened roofing area had high moisture content, probably due to the depolymerization of resin.Mean compressive strength (*σ*_10, mean_) of the extracted degraded samples was 93% lower than the declared value.Microanalyses showed the random distribution of basalt fibers with locally melting binder inserts in the form of droplets. The surface morphology of aged samples showed that fibers were surrounded by a non-uniform layer with many cracks and bulges.Different chemical composition was detected on the surface of aged samples compared to new ones. Aged samples showed a relative increase in Al and Ca, and we might assume that Al_2_O_3_ and CaO deposits were formed on the surface.Heterogeneities near the aged fiber surface on a scale of 10 nm showed the formation of a thin ribbed layer on the surface that might indicate ion exchange on the sample surface.Combustion tests affected the aged sample to a lesser extent because formaldehyde resin had already degraded to some degree before the analysis was carried out.

The results show that humidity can and will inevitably penetrate the mass of insulation applied in roofing. Therefore, it is very important to study how natural aging under high humidity and thermal oscillations affects the durability of insulating building materials. In order to reduce the negative impacts of moisture and increase the durability of insulation materials, new samples with more persistent hydrophobic admixtures could be used along with synthetic binders that have the highest possible degree of curing. Last but not least, proper installation and inspection during roofing installation can significantly contribute to the durability of building materials in many cases.

## Figures and Tables

**Figure 1 materials-13-02392-f001:**
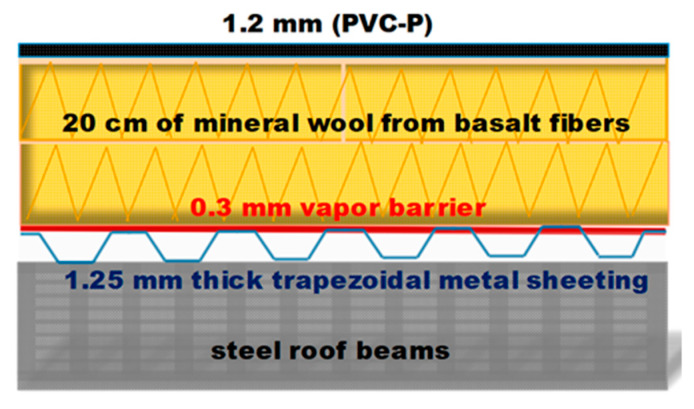
Scheme of the roofing structure.

**Figure 2 materials-13-02392-f002:**
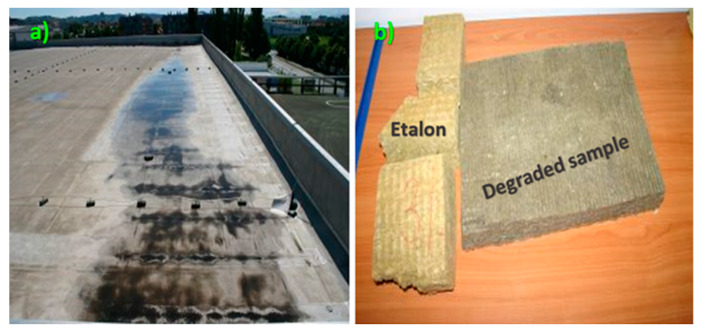
(**a**) Mineral wool samples taken from softened, uneven roofing areas of a commercial building located in Slovenia, Central Europe. (**b**) Visual presentation of new and degraded insulation samples.

**Figure 3 materials-13-02392-f003:**
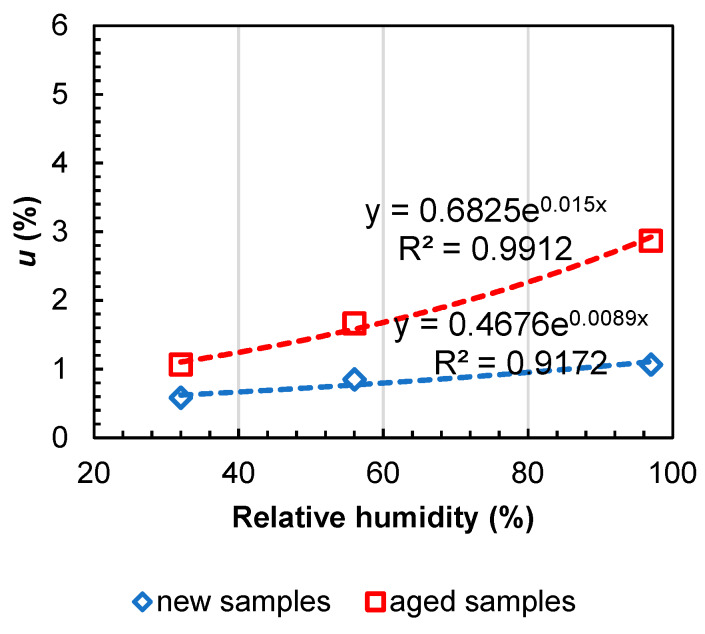
Moisture content (*u*) at different relative humidities: comparison between the new and aged samples.

**Figure 4 materials-13-02392-f004:**
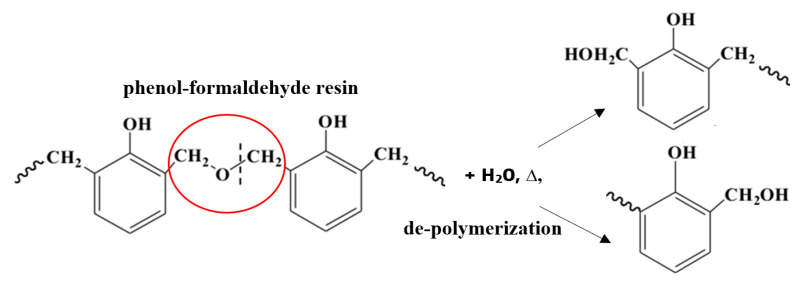
Partial degradation of resin by depolymerization initiated by hydrolysis.

**Figure 5 materials-13-02392-f005:**
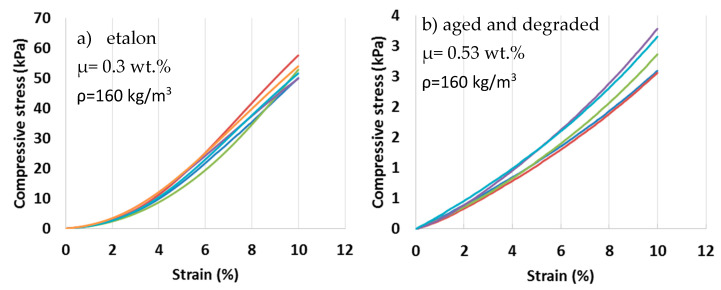
Compressive stress at 10% deformation.

**Figure 6 materials-13-02392-f006:**
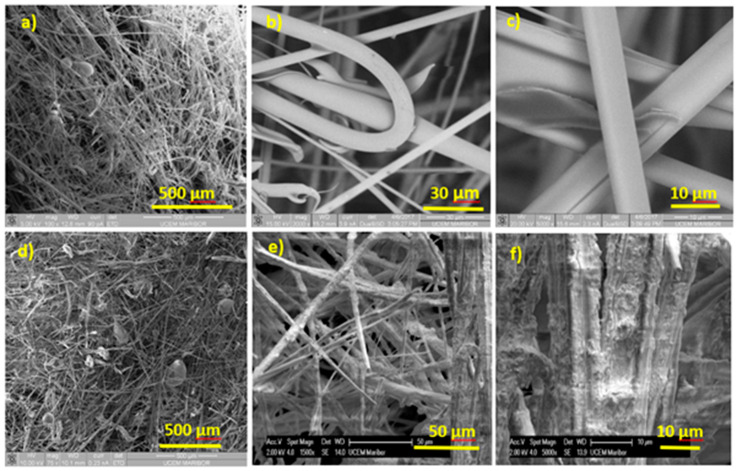
SEM microanalysis of new (**a**–**c**) and aged (**d**–**f**) insulating material.

**Figure 7 materials-13-02392-f007:**
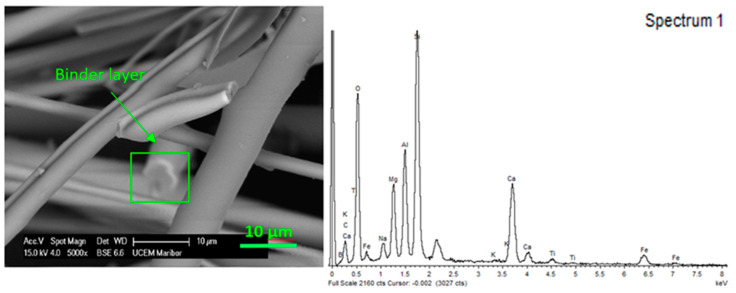
SEM-EDX was used to observe the composition of the surface of the new basalt wool sample coated with binder. The thick white uniform layer that surrounds the basalt fiber is the coated binder.

**Figure 8 materials-13-02392-f008:**
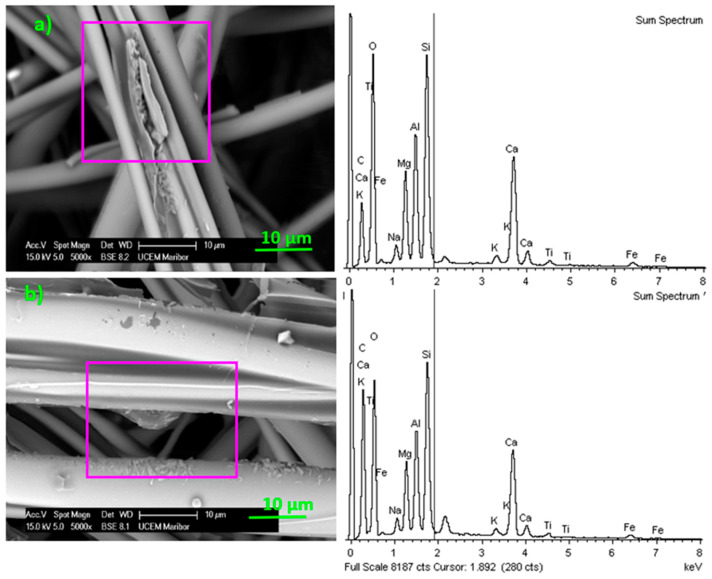
SEM-EDX spectra of the surface of mineral wool fabricated from basalt fibers with two characteristic defects: (**a**) cracks and (**b**) bulges.

**Figure 9 materials-13-02392-f009:**
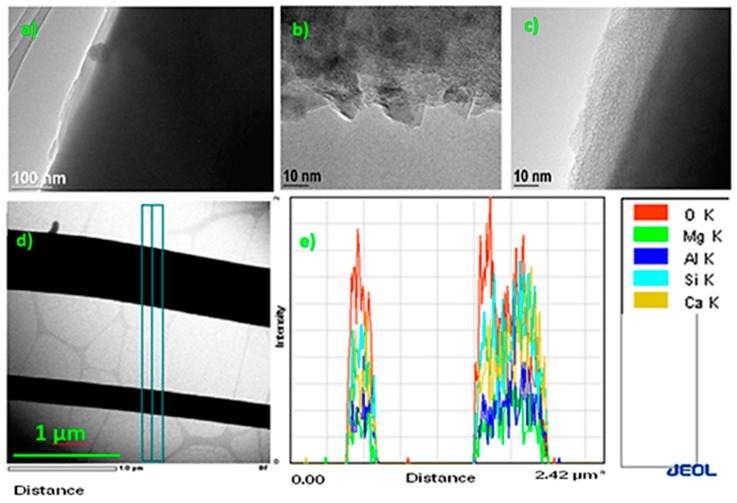
Scanning transmission electron microscope analysis of fiber surface in nano-scale. Visible formation of a thin ribbed layer on the aged basalt surface (**a**–**c**), the parallel cross section of basalt fiber (**d**), and presence of elements on the surface and in the interior (**e**).

**Figure 10 materials-13-02392-f010:**
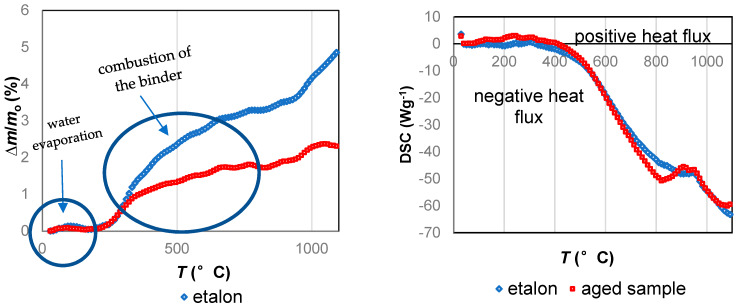
Results of the TGA-DSC analysis between new and aged samples. Normalized mass change **∆***m*/*m*_o_, where **∆***m* and *m*_o_ are the mass change and the initial mass of the sample, respectively.

**Figure 11 materials-13-02392-f011:**
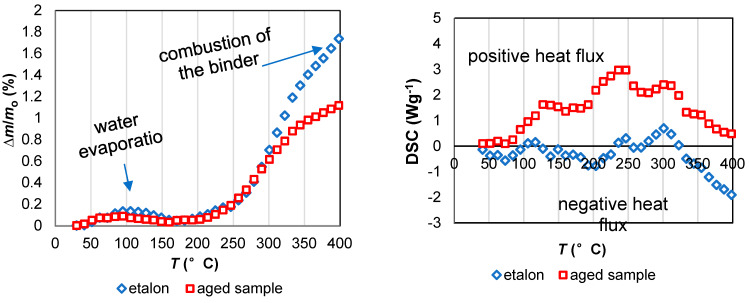
TGA-DSC analysis in the area below the 400°C. Normalized mass change **∆***m*/*m*_o_, where **∆***m* and *m*_o_ are the mass change and the initial mass of the sample, respectively.

**Table 1 materials-13-02392-t001:** Water content (*u*) and compressive strength (*σ*) of degraded mineral wool from basalt fibers.

Sample	*u*_min_ (wt.%)	*u*_mean_ (wt.%)	*u*_max_ (wt.%)	*σ*_10, min_ (kPa)	*σ*_10, mean_ (kPa)	*σ*_10, max_ (kPa)	*σ*_10, dec_ (kPa)
Mineral wool	0.53	0.63	4.90	2.18	5.21	10.20	70

**Table 2 materials-13-02392-t002:** Surface concentration of elements in a new sample (etalon), a sample with crack defects, and a sample with bulge defects (wt.%).

Spectrum	B	Na	Mg	Al	Si	K	Ca	Ti	Fe
Etalon	7.43	0.96	3.26	4.21	9.72	0.09	6.43	0.35	2.45
Cracks	4.22	1.50	4.60	5.96	12.08	0.89	14.11	1.04	2.78
Bulges	4.16	0.92	3.90	5.03	10.24	0.57	10.33	0.77	1.94

**Table 3 materials-13-02392-t003:** Relative amount of elements on the surface compared to Si in new and degraded samples with cracks and bulges (wt.%).

Ratio	B/Si	Na/Si	Mg/Si	Al/Si	K/Si	Ca/Si	Ti/Si	Fe/Si
**Etalon**	0.76	0.10	0.34	0.43	0.01	0.66	0.04	0.25
**Cracks**	0.35	0.12	0.38	0.49	0.07	1.17	0.09	0.23
**Bulges**	0.41	0.09	0.38	0.49	0.06	1.01	0.08	0.19

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
