# Peer review of "The Influences of Moisture on the Mechanical, Morphological and Thermogravimetric Properties of Mineral Wool Made from Basalt Glass Fibers"

_materials, 2020, doi:10.3390/ma13102392_

Round 1

Reviewer 1 Report

In this work, both unused and from the softened roofing area obtained mineral wool were comprehensively analyzed in a laboratory using different characterization techniques. The results show that natural aging under high humidity and thermal fluctuations greatly affects the surface morphology and chemical composition of fibrous composite. The research topic is interesting and meaningful. However, some errors and deficiencies were also discovered. Therefore, I think this paper can be published in Materials after a minor revision.

  1. Backgrounds need to be strengthened. Following references were suggested to be added in the Introduction.
  • Construction and Building Materials, 2014, 55, 220-226;
  • Journal of materials in civil engineering, 2017, 29, 04016225;
  1. Please provide a schematic diagram of the several layers that make up the roofing
  2. The authors used at least 5 sample to text the moisture content and mechanical property of the degraded mineral wool samples. However, the difference between umin (0.53) and u max (4.90) is nearly 10 times, and u mean value (0.63) is close to u min (0.53). So, is the value of u max representative? Should more samples be selected for testing?
  3. Number the illustrations in Fig. 7 according to their sequence in the text. And please provided a more detailed caption.
  4. Page 9, “Weight loss between 30°C and 150°C corresponds to the evaporation of water content.”

Q: In this case, why test the moisture content in 105°C instead of 150°C?

  1. In the conclusion, the author only pointed out that humidity can and will inevitably penetrate the mass of insulation applied in roofing, but did not propose any improvement methods. 

Author Response

Sincerely,

The Authors

Reviewer 2 Report

The reviewed manuscript titled: “The influences of moisture on mechanical, morphological and thermogravimetric properties of mineral wool made from basalt glass fibres” requires a major correction of a native speaker (English). Also readability of some figures are not sufficient (Figure 6, 7). In the paper there is no information about novelty of proposed method of investigation or used material (mineral wool made from basalt glass fibre). The paper is well written and the development of the proposed formulation is consistent. Despite this the paper is well written and the development of the proposed formulation is consistent. The reviewed article can be published in Materials Journal after minor corrections. All necessary corrections are marked in attached file.

Author Response

Sincerely,

The Authors

Reviewer 3 Report

Comments provided in attachment

Author Response

Sincerely,

The Authors

Reviewer 4 Report

“The influences of moisture on the mechanical, morphological, and thermogravimetric properties of  mineral wool made from basalt glass fibers” investigates on the degradation of mineral wool from basalt fibers 

used as thermal insulation material, due to the natural aging under high humidity and thermal fluctuations exposition.

Revision or comment

The work is interesting and presents several experimental analyses to give an accurate scene about natural aging of a material frequently used as insulating material in construction systems. However, more details about method used for the analyses of the samples studied are necessary. A clearer description can make more robust results.   

It is not clear how many samples have been analysed. Table 1 reports different values, u min, mean and Max and for each of those values figure 3 shows different traces. What are those traces? Different samples for minimum values and so on… or are they different measurements on the same sample? A standard deviation on the mean is useful to understand variability on measurements.

The higher moisture content samples are an exceptions or not?

Author Response

Sincerely,

The Authors

Round 2

Reviewer 3 Report

Please refer to the attachment for comments. 
